# The casts of Pompeii: Post-depositional methodological insights

**Llorenç Alapont**[1], **Gianni Gallello**[1]*, **Marcos Martinón-Torres**[2], **Massimo Osanna**[3], **Valeria Amoretti**[4], **Simon Chenery**[5], **Mirco Ramacciotti**[1,6], **José Luis Jiménez**[1], **Ángel Morales Rubio**[6], **M. Luisa Cervera**[6], **Agustín Pastor**[6]

1 Department of Prehistory, Archaeology and Ancient History, University of Valencia, Valencia, Spain, 2 McDonald Institute for Archaeological Research, University of Cambridge, Cambridge, United Kingdom, 3 Ministry of Cultural Heritage and Activities and Tourism, Rome, Italy, 4 Archaeological Park of Pompeii, Naples, Italy, 5 British Geological Survey, Environmental Science Centre, Keyworth, Nottingham, United Kingdom, 6 Analytical Chemistry Department, Edifici Jeroni Muñoz, University of Valencia, Burjassot, Spain

* gianni.gallello@uv.es

**Data Availability Statement:** All relevant data are within the paper and its Supporting Information files.

## Abstract

The casts of Pompeii bear witness to the people who died during the Vesuvius 79 AD eruption. However, studies on the cause of death of these victims have not been conclusive. A previous important step is the understanding of the post-depositional processes and the impact of the plaster in bones, two issues that have not been previously evaluated. Here we report on the anthropological and the first chemical data obtained from the study of six casts from Porta Nola area and one from Terme Suburbane. A non-invasive chemical analysis by portable X-ray fluorescence was employed for the first time on these casts of Pompeii to determine the elemental composition of the bones and the plaster. Elemental profiles were determined providing important data that cross-referenced with anthropological and stratigraphic results, are clearly helpful in the reconstruction of the perimortem and post-mortem events concerning the history of these individuals. The comparative analyses carried out on the bone casts and other collections from burned bones of the necropolis of Porta Nola in Pompeii and Rome Sepolcreto Ostiense, and buried bones from Valencia (Spain), reveal the extent of high temperature alteration and post-depositional plaster contamination. These factors make bioarchaeological analyses difficult but still allow us to support asphyxia as the likely cause of death.

## Introduction

The casts from Pompeii (Naples, Italy) represent one of the most important remains of the Vesuvian town; they are the expression of a dramatic event that made Pompeii an exceptional site, providing a focus of Roman culture studies [1]. Obtained in the 1870s by pouring plaster in the voids left by bodies that had decomposed under calcified volcanic ash, the casts typically contain skeletal remains embedded in a plaster cast that retains the body shape. These remains are for the most part human, giving a realistic and moving image of the people who lived and died in Pompeii, victims of the Vesuvian eruption of 79 AD. Several works have focused on identifying the cause of death of these individuals. The main contrasting hypotheses developed

**Funding:** This study was funded by Ministerio de Universidades, BEAGAL18/00110, Dr Gianni Gallello, Conselleria de Innovación, Universidades, Ciencia y Sociedad Digital, Generalitat Valenciana, PROMETEO 2019-056, Dr M. Luisa Cervera, Horizon 2020 Framework Programme, H2020-MSCA-IF-2015-704709-MATRIX., Dr Gianni Gallello. The funders had no role in study design, data collection and analysis, decision to publish, or preparation of the manuscript.

**Competing interests:** The authors have declared that no competing interests exist.

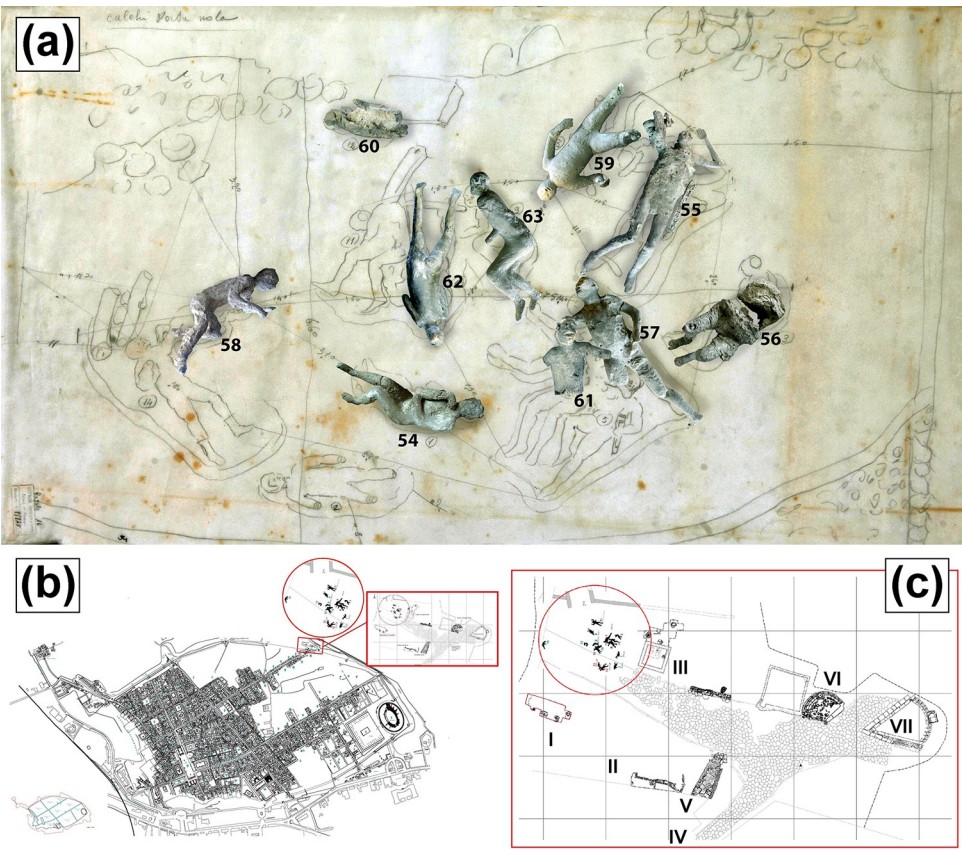

**Fig 1. Location of casts from Porta Nola.** (a) Original position of some of the studied Porta Nola casts (#57, #62, #58, #54, #55). (b) Map of Pompeii. (c) Detail of casts discovery area (I: Burials of Praetorians; II: Modern Masonries; III: Tomb of Obellius Firmus; IV: Porta Nola; V: Leakpan; VI: Tomb of Esquilia Polla; VII: Anonymous Tomb).

were either asphyxia [2, 3] or body evaporation supported by Petrone et al. (2018) [4], and dehydration, suggested by Martyn et al. (2020) [5]. Both studies were based on the Herculaneum population [4, 5]. Here we report the first data from a multidisciplinary study carried out on seven Pompeii cast remains: six fugitives from Porta Nola area (Fig 1) and one from Terme Suburbane, cross-referencing the obtained anthropological and chemical results with studies [2, 6, 7] of stratigraphic level that show the pyroclastic succession and the sequential events to better define the cause of death of the studied Pompeii individuals.

Petrone et al. studied several skeletons found in the waterfront chambers, also called *fornici*, along the Herculaneum beach where some individuals looked for shelter during the eruption. Those authors propose that these individuals died instantly during later pyroclastic flows due to the vaporization of body fluids and demonstrate an "incomplete pugilistic attitude" from temperatures of more than 408°C [4]. On the other hand, studies such as the one of Martyn et al., also carried out on some Herculaneum *fornici* bone remains, suggested that the pyroclastic surges were below 400°C and provoked a non-instantaneous dehydration, as indicated by the muscle contraction, "boxer position", and the preservation of collagen in the studied skeletal remains [5]. A death from asphyxia caused by ashes was also supported by different studies carried out on individuals from Pompeii [2, 3]. To confirm this last hypothesis, recently, some authors calculated the pyroclastic density current. Described as "a ground hugging gas-particle flows" it was originated from the eruption column collapse or the fall of the lava dome. Then it spread out in Pompeii at low temperature over about 17 min. This combined with ash particles

(smaller than 0.001 mm), gave enough time to cause death by asphyxia at Pompeii. As suggested by Dellino et al. (2021) [2], the human exposure to fine ash, even at a low particle concentration, can be tolerated for only a few minutes. This research conducted a bioarchaeological approach on seven Pompeii casts. Furthermore for the first time non-invasive chemical analysis by portable X-ray fluorescence (pXRF) was carried out on these casts to determine the elemental composition in the bones and the plaster. This analytical technique has been already employed in skeletal remains to clarify taphonomic and post-depositional issues [8, 9] and in osteoarchaeological and forensic studies to identify individuals and pathologies [10–14].

## Methodology

### Test samples

Six casts (#57, #62, #58, #64, #54, #55) from Porta Nola area and one from Terme Suburbane (#34), were selected for anthropological and multielemental analysis. The methodological approach developed in the casts analysis included the pXRF measurements of pre-eruption cremated bones, excavated by one of us (L.A.), from the same area (Porta Nola), where the casts were found, to compare the results and control in situ contamination and post-depositional factors related to the elemental profile. These remains were compared with cremated bones from another Roman period necropolis (also excavated by L.A.) located in Rome. Finally, these two groups were compared with buried bones from a Spanish Islamic necropolis. In all the cases, the surface of the cast bones, and the other cremated and buried bones were previously cleaned by employing a wet cloth with distilled water to avoid dust contamination. In total, 20 cremated bone fragments from Pompeii (CR. Po.) from different urns excavated in the necropolis of Porta Nola (fieldworks campaigns from 2015 to 2017) [15], 24 cremated bone fragments (CR. Ro.) from the Necropolis of Rome Sepolcreto Ostiense (fieldwork campaign 2018) [16] and 11 buried bone fragments from the Islamic necropolis of Colata (fieldworks campaign 2003) [17], located in the heart of the Vall d'Albaida (Montaverner, Valencia, Spain) (BR. Al.) were sampled.

### Anthropological study

The biological sex was estimated employing the pelvis [18, 19], the structural differences of the skulls [20], the measurement of the diameter of the femoral head and humerus [21], the length of the glenoid cavity of the scapula [20] and measurements of the calcaneus using a discriminant function [22]. Furthermore a probabilistic sex diagnosis method [23], based on metric data was applied and a composite morphological method taking into account five morphological criteria on the coxal bone [24] was employed. To differentiate between young and older adults the degree of fusion of the medial ossification centre of the clavicle and the degree of fusion of the iliac crest to the iliac bone [25] were taken into account. Further assessment of age-at-death were based on the remodelling degree of both the auricular surface of the coxal bone [26] and pubic symphysis [27]. Anatomic variation, degenerative and osteoarticular pathologies, traumas, oral pathologies including dental wear were observed according to the methods proposed by Mann and Murphy [28] and Bass [20].

### pXRF

The pXRF was employed to identify the elemental composition of the analysed bones and plaster. X-ray fluorescence spectra were obtained in situ by applying the device, a portable S1Titan X-ray fluorescence spectrometer by Bruker (Kennewick, Washington, USA) equipped with a Rhodium X-ray tube and X-Flash® SDD detector on the surface of the measured points of the

casts and reference collections. The manufacturer provided Geochem-trace programme was employed to quantitatively determine the elemental content of Al, Si, K, Ca, P, S, Cl, Ti, Mn, Fe, Zn, Sr and Pb. The analytical technique employed in this work was adequately validated, employing standard reference and sample control materials, in order to obtain accurate results. The certified material Bone Ash NIST 1400 was used as a standard reference to evaluate the accuracy of the obtained data for K, Ca, P, Fe, Zn, Sr and Pb. To control the reliability of the other elements (Al, Si, Cl, Ti and Mn), the same matrix (bone) was employed. All the samples including the certified material were scanned for one minute. Each sample was analysed three times and the average concentrations were taken into account. The quality control of the results demonstrated that all the obtained elemental results have less than 5% error compared to the certified results.

## Data processing and statistics

Data analysis was performed in R (version: 4.1.2) [29]. Principal component analysis (PCA) was employed to explore the dataset. PCA was performed in order to observe similarities among buried, cremated and cast bones, and plaster samples. For this, the classes including the casts (Cast), plaster (Plaster), and the reference collections such as the cremated bones from Pompeii (CR. Pompeii) and Rome (CR. Rome) and buried bones from the Islamic necropolis of Vall d'Albaida (BR. Albaida) have been created to run the statistical analyses. All the analysed elements (Al, Si, K, Ca, P, S, Cl, Ti, Mn, Fe, Zn, Sr and Pb), and the ratios Ca/P and Sr/Ca were used as variables. Data were autoscaled prior to the PCA. Finally, plots were made for Al, Si, K, Ca, P, S, Cl, Ti, Mn, Fe, Zn, Sr and Pb concentrations and Ca/P and Sr/Ca ratios. The ratio Ca/P have been used to control the integrity of the mineral component of bone [30]. The fresh bone has an estimated Ca/P ratio ranging between 2.17 and 2.31 [31]. Sr/Ca in archaeological bone, is highly susceptible to diagenesis [32]. In this work both ratios have been specifically employed to evaluate post-depositional processes comparing reference collections of burned and buried bones with bone casts and plaster.

The following R packages were used for data analysis and visualisation: factoextra (version: 1.0.7) [33] and ggplot2 (version: 3.3.5) [34].

## Results

### Bioarchaeological analysis on casts

The cast #57 (Fig 2) is an adult individual (25–30 years old). He was found supine and has some flexed or lopsided anatomical areas. The skull is inclined to the right and the chest is straight, presenting its coxofemoral joint slightly rotated towards its medial aspect.

The cast #62 (Fig 3) is an adult individual (25–30 years old) identified as female, found interlaced to the left hand-forearm and holding a rope that would go over the left shoulder carrying a sack or bag. The original position of this individual was prone with the head resting on the face.

The cast #58 (Fig 4) is an adult individual (20–25 years old) identified as a male. This was divided into 7 parts before its restoration. This cast appears to preserve the morphology of the right ear.

The cast #64 (Fig 5) is a male adult individual between 45 and 50 years old, very altered and remodelled in an old restoration, especially in the zygomatic-facial area. This individual was originally found supine, with the upper extremities separated from the body.

The cast #54 (Fig 6) was identified as a female adult (35–40 years). This individual was found in a left lateral decubitus position, with the head resting on the left temporal parietal, the right arm with the elbow fully flexed and the forearm next to the chest.

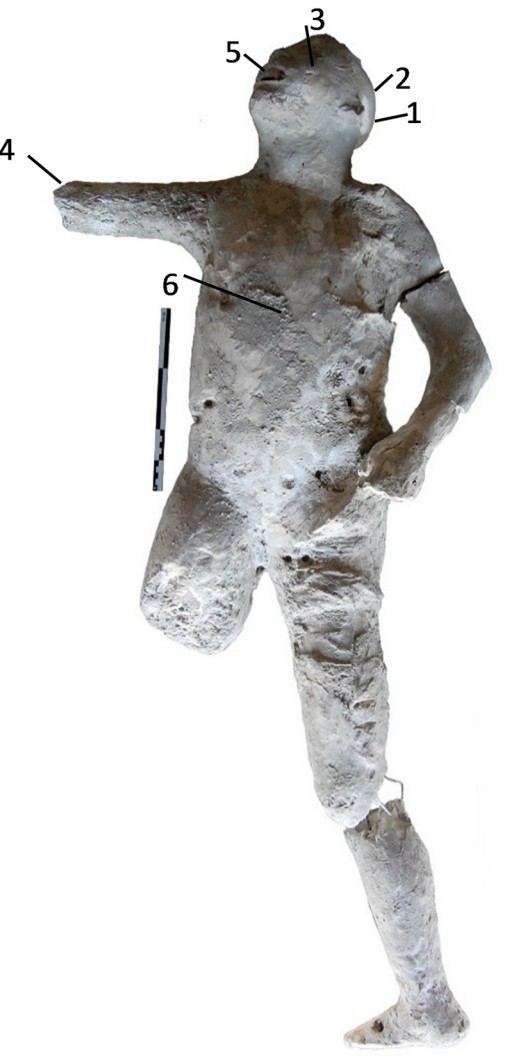
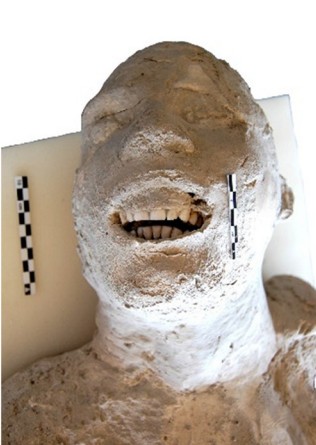

**Fig 2. Cast #57 and pXRF measuring points (1–6).**

The cast #55 (Fig 7) belongs to a male adult between 25 and 30 years old, originally in supine position, with the upper extremities separated from the body, and the forearms and hands on the head.

The cast #34 (Fig 8) from Terme Suburbane is a male adult individual (20–25 years old). The original position of the individual was right lateral decubitus. The right arm had the elbow flexed, separated from the body. The legs were parallel following the longitudinal axis of the body.

The detailed description of the studied casts can be found in S1 File.

In addition, the paleopathological data showed interesting features in some of the studied individuals. The cast #57 shows dental wear of the incisors and canines of the upper jaw, product of its axial shaft, and erosion of the lower incisors. Individual #58 also presents hypoplasia in the left lateral incisor and dental wear on the lower left lateral incisor, and also an enthesophyte of the left patella while cast #64 presents osteoarthritis in the right knee and in the medial

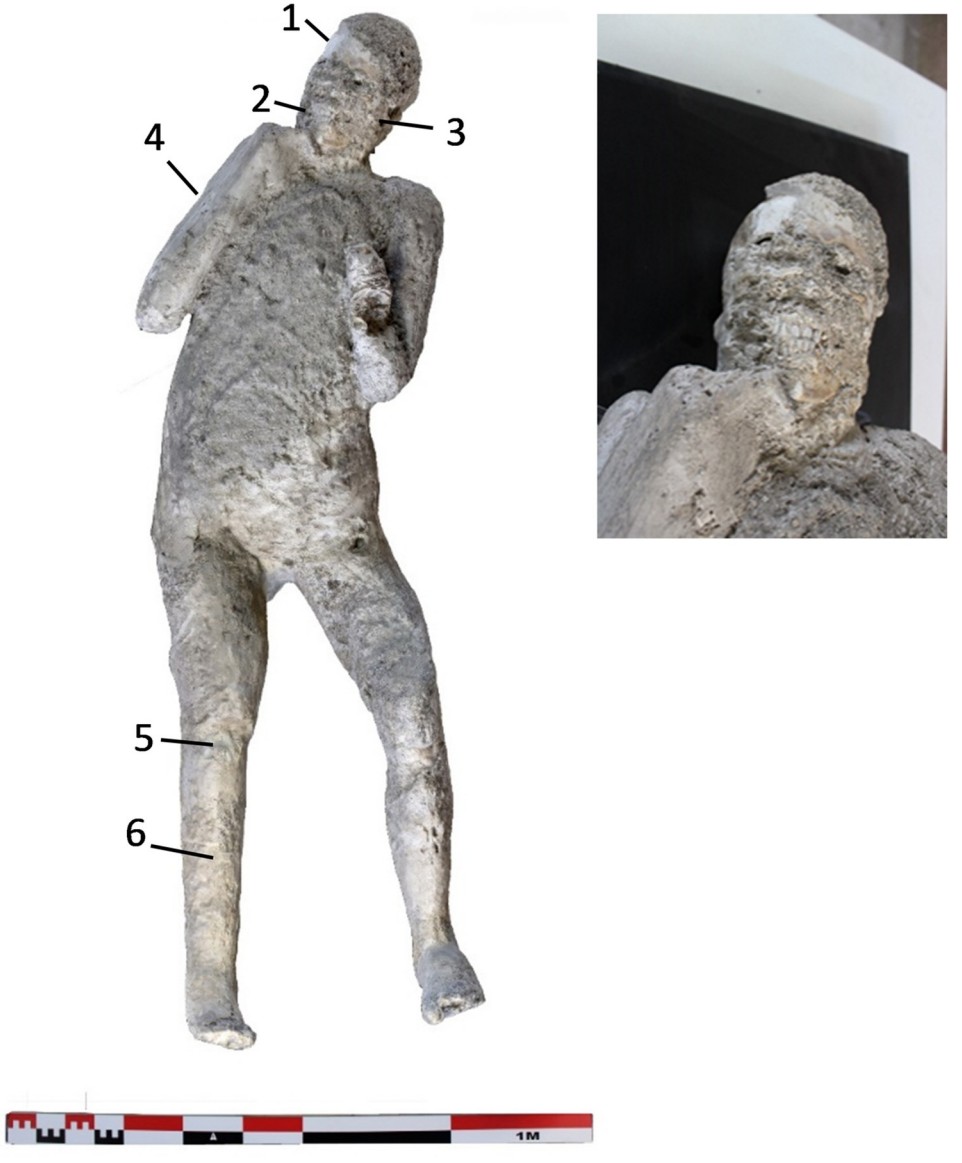

**Fig 3. Cast #62 and pXRF measuring points (1–6).**

condyle of left femur. The radiological examination also revealed traumatic perimortem fractures in the tibia and fibula of the right leg of the cast #58 and in the right humerus of the cast #55. These fractures appear to have occurred close to the time of death of the individuals. No obvious pathological conditions in the individuals #62 and #54 from Porta Nola and #34 from Terme Suburbane, were observed.

## pXRF analysis

Multielement analysis results (see the measuring points in Figs 2–8) are shown in the S2 File and can be visualized in the S3 File. Excepting Ca and P and their ratios that are reported in the text (Figs 9 and 10).

Both cast bone and plaster samples have elemental levels similar to bones for Pb, Mn and Al (S2 File). The different groups of samples also show similar amounts of Zn, although some

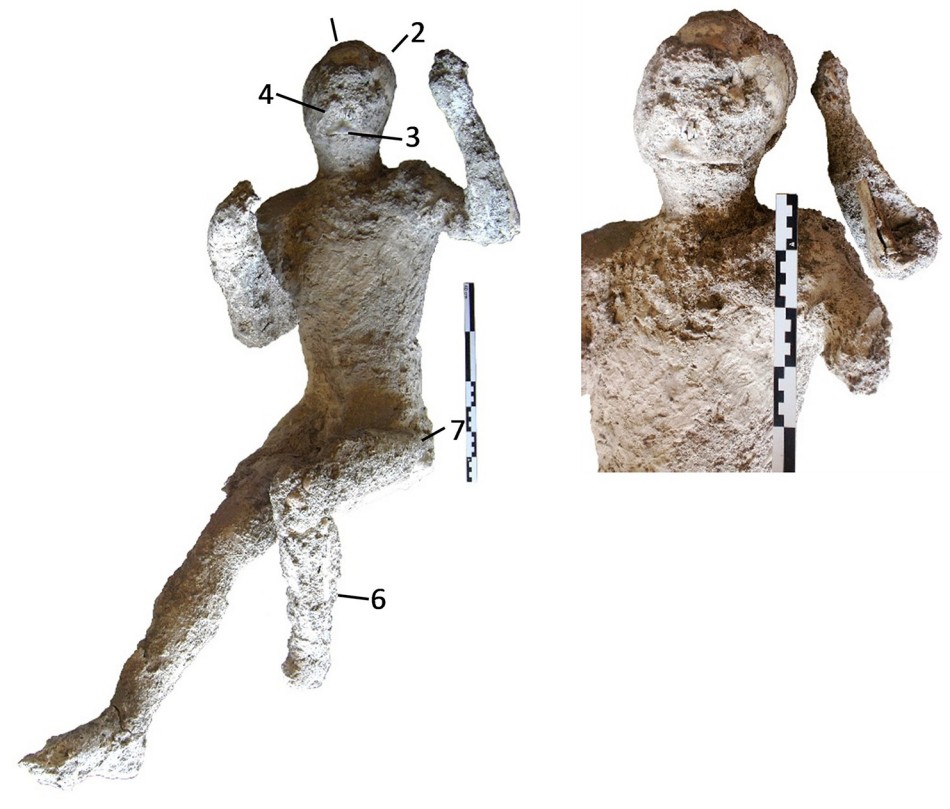

**Fig 4. Cast #58 and pXRF measuring points (1–6).**

casts are characterized by higher amounts of this element, especially in the analyzed teeth of cast #57 and cast #62 (TT). On the other hand, bones are richer in Ca and P (Fig 9) than plaster samples, and buried bones (BR. Albaida) show the highest mean concentrations for these elements. Several cast measured points fall within a standard deviation of the two cremated bones groups and it should be pointed out that cast #34 has the lowest concentrations for both elements (Ca and P). Most of the plaster samples show high concentrations of K, Cl, Ti and Si, and several casts show higher concentrations than bones, although cremated bones from Rome are characterized by higher elemental levels than those from Pompeii and of the inhumated ones. Plaster samples have the highest amounts of Fe, Sr and S. Casts are characterized by levels of Fe similar to those of cremated bones, while, in most of the cases, they have levels of Sr and S higher than the former ones. Concerning the calculated ratios, plasters are enriched in Ca over P (Fig 10A and 10B) and in Sr over Ca (Fig 10C), which is reflected also in some of the studied casts, such as cast #34.

## Principal component analysis (PCA)

The principal component analysis (PCA) of the chemical data was carried out to observe variation of cast bones with respect to the comparative sets of burned bones, buried bones and plasters (Fig 11). The first two PCs explain 63.7% of the overall variance in the dataset.

As can be observed in the samples/scores plot (Fig 11A), plaster samples have higher PC1 scores than bone samples. It is worth noticing that samples of buried bone group together in

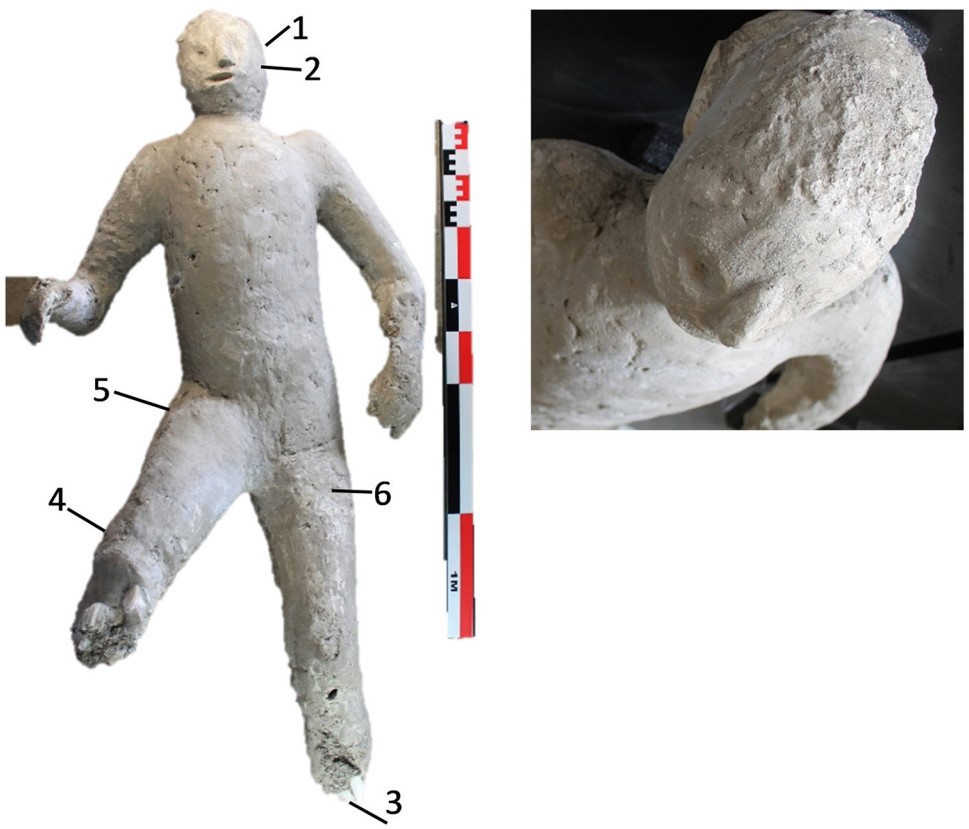

**Fig 5. Cast #64 and pXRF measuring points (1–6).**

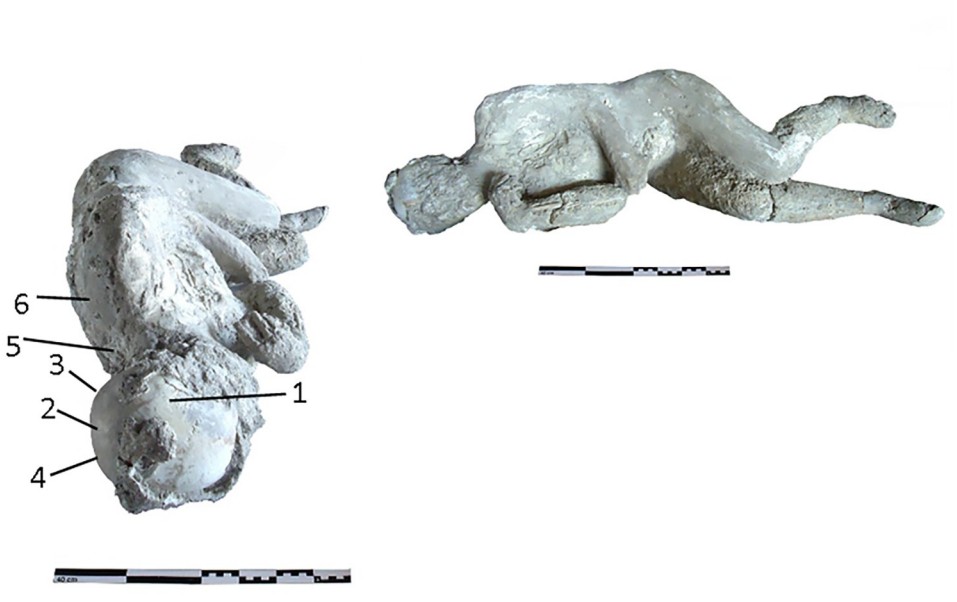

**Fig 6. Cast #54 and pXRF measuring points (1–6).**

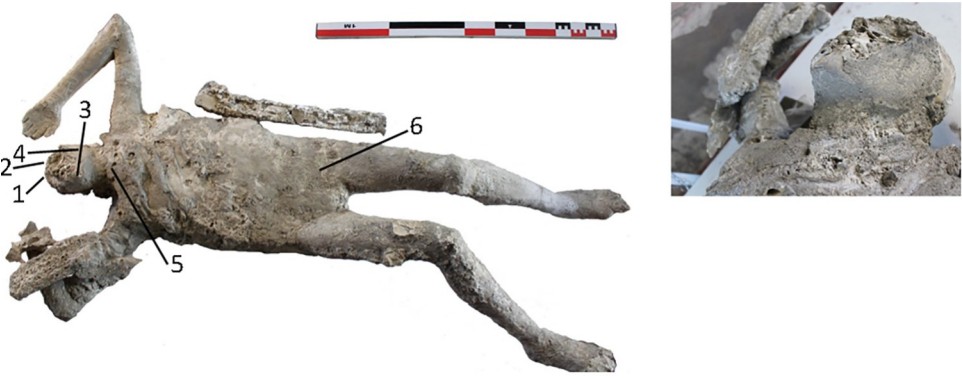

**Fig 7. Cast #55 and pXRF measuring points (1–6).**

the third quadrant of the diagram, while those from cremated bones are scattered on both PC1 and PC2. Most of the cast samples fall within cremated bones. The colored small squares and circles referring to cremated bone colors do not show a clear grouping although the majority of the brownish bones are similar to the buried ones, which is consistent with their exposure to

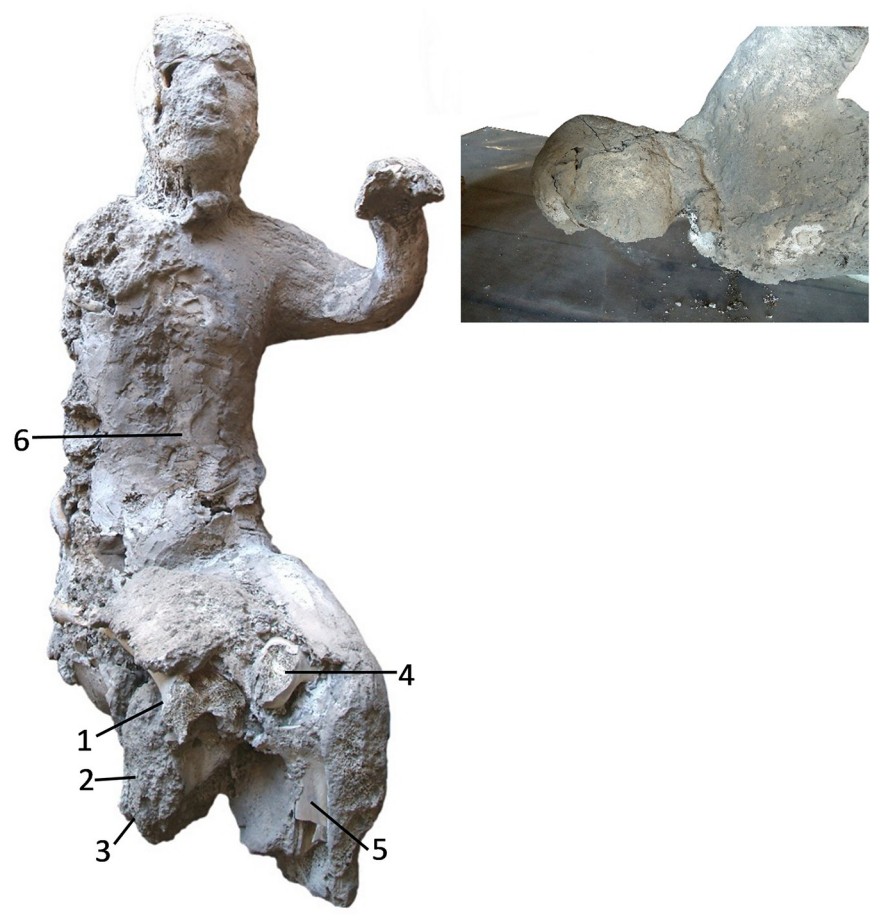

**Fig 8. Cast #34 and pXRF measuring points (1–6).**

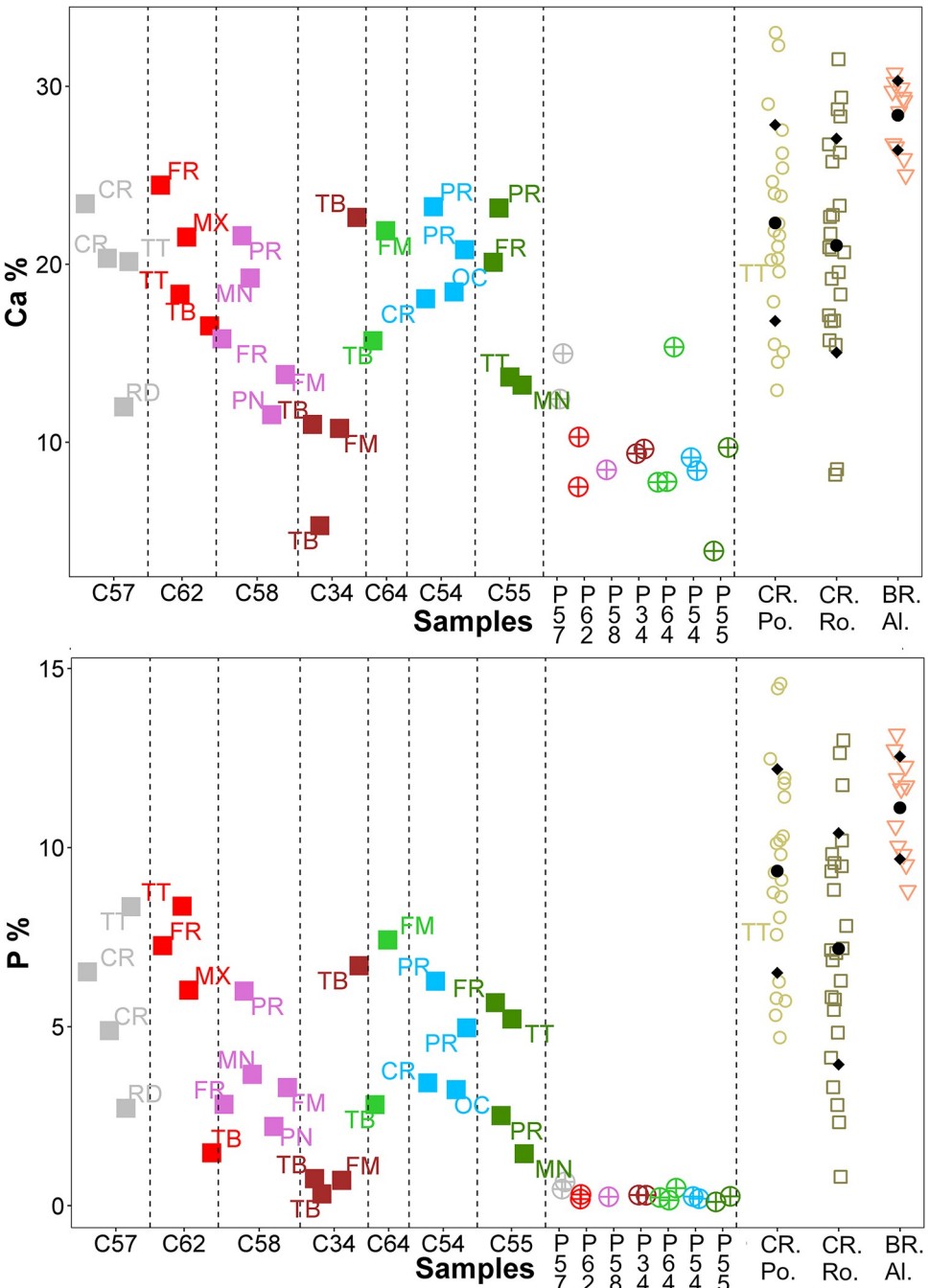

**Fig 9. Ca and P concentrations for all the analysed materials.** Cast bones (C57, C62, C58, C34, C64, C54, C55) and plaster (P57, P62, P58, P34, P64, P54, P55), cremated bones from Pompeii (CR. Po.) and Rome (CR. Ro.), and buried bones from the Islamic necropolis of Vall d' Albaida (BR. Al.). In the graph, for each cast the kind of bone is indicated: cranium (CR), parietal (PR), occipital (OC), frontal (FR), maxilla (MX), mandibular (MD), tooth (TT), radius (RD), peroneal bone (PR), tibia (TB), femur (FM). •: mean; ◆: one standard deviation.

lower temperatures. Concerning cast samples, the majority of the cast bones are located close to cremated bones, although some fall between bones and plaster (C34.1, C34.4, C55.5, C57.4) or close to plaster (C34.3), probably due to the results of being encapsulated by plaster (Fig 11A). Variables/loadings plots (Fig 11B) indicate that scores for PC1 (50.1%) are mainly driven

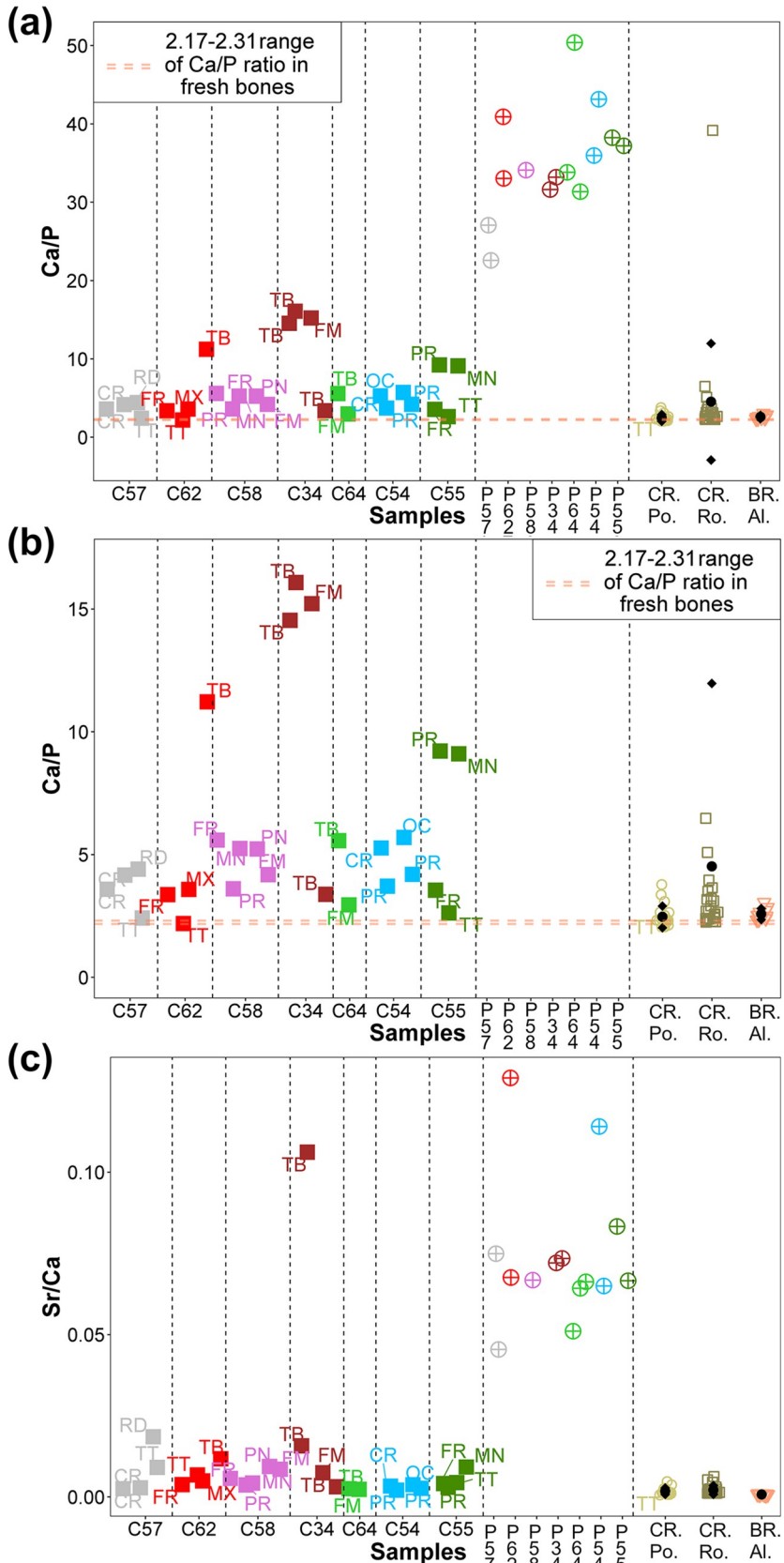

**Fig 10. Ca/P ratio (a) Diagram for all the analyzed materials.** (b) Diagram excluding plaster (in the range up to 20) and Sr/Ca for all the analysed materials (c). Cast bones (C57, C62, C58, C34, C64, C54, C55) and plaster (P57, P62, P58, P34, P64, P54, P55), cremated bones from Pompeii (CR. Po.) and Rome (CR. Ro.) and buried bones from the Islamic necropolis of Vall d'Albaida (BR. Al.). In the graph for each cast (Cast) the kind of bone is indicated: cranium (CR), parietal (PR), occipital (OC), frontal (FR), maxilla (MX), mandibular (MD), tooth (TT), radius (RD), peroneal bone (PR), tibia (TB), femur (FM). •: mean; ♦: one standard deviation. Orange line shows the interval ratio (2.06–2.45) in fresh bones.

by P and Ca concentrations in the negative direction and by most of the other elements and Ca/P and Sr/Ca ratios in the positive one. For PC2, S, Sr and ratios are the most influential variables in the negative direction, and Al, Si, Ti, Mn, Zn and Pb in the positive one. The second PC (13.6%) provides little information to differentiate among the classes ([Fig 11A]).

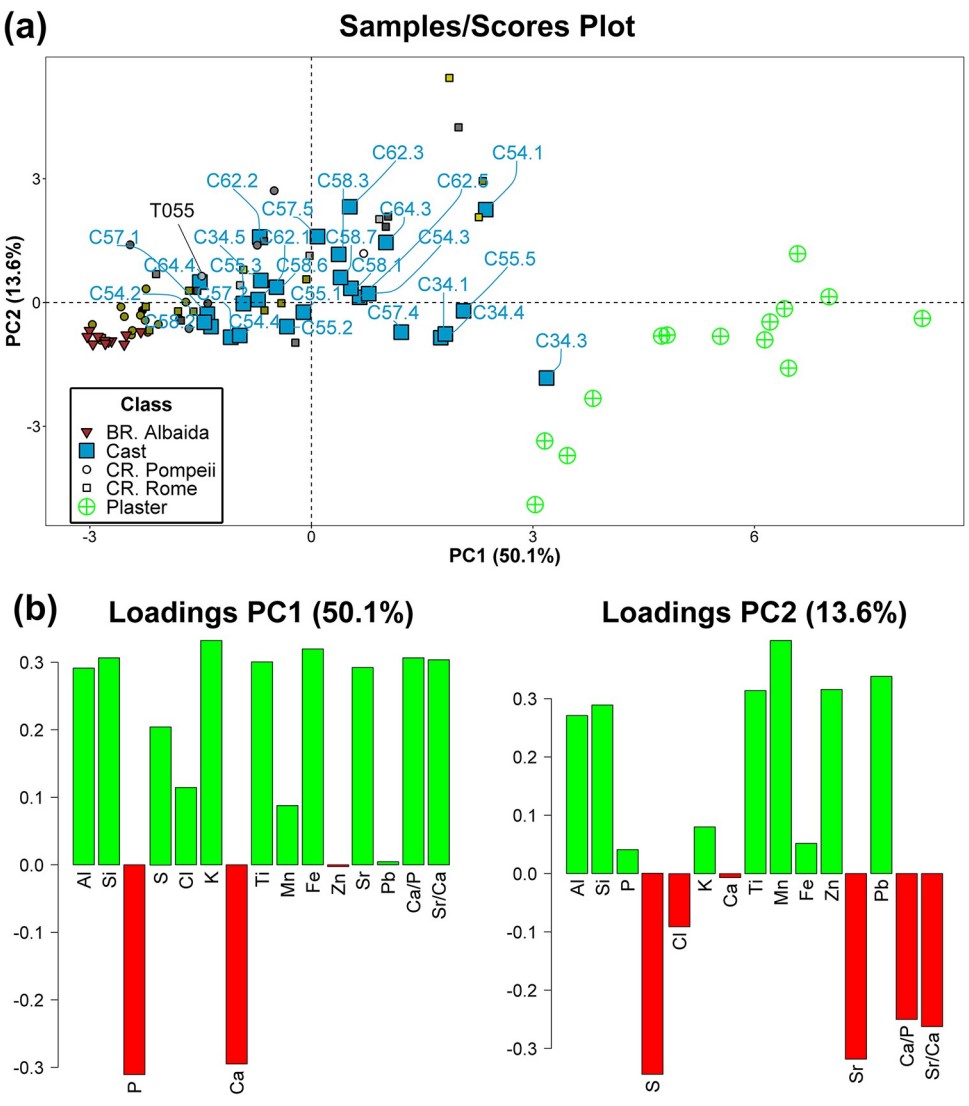

**Fig 11. Principal component analysis of chemical data from Porta Nola casts and test samples.** PCA scores plot with the labeled casts measured points. (a) Samples/scores plot for cast (Cast), plaster (Plaster) and cremated bones from Pompeii (CR. Pompeii) and Rome (CR. Rome) and buried bones from the Islamic necropolis of Vall d' Albaida (BR. Albaida). Colored cremated sample points refer to bone color. (b) Variables/loadings plots for PC1 and PC2.

## Discussion

### The elemental data

If we look at the elemental data, the obtained results show that P and Ca concentrations are the most important variables in the negative direction of PC1, being particularly high in buried bones (Fig 11B). The individual #34 suffered major plaster contamination in tibiae (C34.1 and C34.3) and femur (C34.4), as confirmed by Ca, P and Ca/P ratio graphs (Figs 9, 10A and 10B), while the parietal of the same individual (C34.5) seems to suffer only minor plaster contamination. Ca/P ratio, calculated from Hancock et al. [31] for fresh bones, was typically between 2.17 and 2.31, values close to the bone biological range will be indicative of the concurrence of both dissolution and recrystallization processes, values above this range will be indicative of processes of precipitation of new authigenic phases (Fig 10A and 10B) [35]. The mechanisms behind these processes were further explained by Nielsen-Marsh et al. [36]. The bone surface apatite is highly reactive and after death the biologically formed minerals become unstable and susceptible to the burial environment. Therefore, $Ca^{2+}$ and $PO_4^{3-}$ ions are exposed and can be totally or partially replaced by the ions of other elements such as the Sr in case of the Ca. The three analysed cast teeth C57.5, C62.2 and C55.3, and the Pompeii burned tooth T055 show a Ca/P ratio in the interval of fresh bones suggesting an insignificant plaster contamination (Fig 10A and 10B). More interesting, in PCA (Fig 11) can be observed that the three teeth have a chemical profile similar to the burned bones, including the tooth T055, which indicates a proportional decrease of Ca and P in these samples. No clear differences were observed among casts, plaster, cremated and buried bones (S2 File) for the other elements.

Thus, the use of plaster as a consolidant significantly affected the elemental profiles of some analysed cast bones, while at the same time it helped preserve important information such as the identification of the perimortem position and the presence of objects as garments.

### The cause of the death and the post-mortem process

A key question of this study is how our multielemental and anthropological analysis combined together can provide data to test hypotheses about the cause of death of Porta Nola fugitives. Some of the Pompeii bodies were found under more than three meters of lapilli, buried under the building collapses and blocked by pumice, while those who were outside died under the collapse of the houses, devastated by the rocks and pyroclastic material expelled by the volcano. This likely happened during the first few hours of eruption as suggested by the stratigraphic studies of Luongo et al. [6]. Nevertheless, the stratigraphic position of the studied Porta Nola fugitives indicates that they survived the early catastrophic event and died 20 hours later from the start of the eruption after the pyroclastic deposition of lapilli [7].

Focusing on the victims of Porta Nola, the stratigraphic data [6, 37] suggest that these people were fleeing the city when the rain of lapilli ended. Walking on the layer of pumice would have been quite difficult and some of them employed branches as a walking stick (see cast #54); after the first phase of the eruption these people were overcome by the dilute pyroclastic density currents of the final phase which were caused by the eruptive column collapse. The passage of these currents lasted for several minutes and even though the temperature of the gas-ash mixture was not very high, the individuals could not breathe this ash for long, resulting in asphyxia [2]. There are many studies on the devastating effects of volcanic ash and gases in the body; when gasses are inhaled and the ash enters in the respiratory system, breathing becomes impossible [38]. These problems would have been compounded by collapsed buildings and the fallen trees as evidenced by various casts of tree trunks in the Porta Nola area.

The victims of Porta Nola do not show a "pugilistic" position and they are not carrying out any action or movement despite their attempts to escape. All of them appear to be lying on their back or prone or on their side, in a relaxed position; some of them covering themselves with garments. This position suggests that ashes and volcanic gases caused the death of the exhausted and asphyxiated fugitives of Porta Nola.

Nonetheless, we should take into account that the deposition from the pyroclastic current formed a layer of hot ash that had thermal impact on corpse of people that have already died of asphyxia and were lying on the ground. The elemental data are consistent with the hypothesised thermal phenomena, corroborated also by ash stratigraphy and taphonomy. Our results show that the analysed cast bones negligibly affected by plaster contamination (see the aforementioned Ca/P ratio in teeth) have a similar chemical profile to cremated individuals from necropolises in Pompeii (pre-eruption necropolis) and in Rome, and different from buried bones (Vall d'Albaida necropolis). The higher levels of Ca and P in buried bones compared to cremated ones can be explained by the thermal impact suffered by bones, which induces both a chemical and mineralogical change in the bones with the consequence of a major leaching of these two elements. Probably the thermal impact together with post-depositional processes allowed the leaching of carbonates and phosphates, and this could explain the lower Ca and P concentrations in burned bones [39, 40]. The reconstruction of the aforementioned elemental behaviour together with the presence of textiles fingerprinted by the plaster maybe support the hypothesis of an exposure of the studied remains to high temperature in a post-depositional environment. The process can be described following the stratigraphy of the deposits of Sigurdsson et al. [7]: first, the individuals suffered asphyxia and were killed by fine ashes while they were laying on the ground trying to cover themselves with clothes, and with the fine ashes taking the shape of surrounding objects, including fine textiles. There are many examples of garments and objects such as a tied sack or objects made by leather or hemp preserved during the burial by ashes, which were identified in the casts of Porta Nola through the plaster imprint. Similarly, textile impressions have been observed in several of the studied casts (casts #56, #57, #34, #55 and #58). In some cases, fine fabrics were observable (cast #57, Fig 12), while in other cases different textiles, most likely made of wool, were identified (casts #34 and #56). Later, the dead bodies were covered by the ground surge materials. Finally, the bodies under the high temperature (over 250˚C) of the ashes left by pyroclastic current suffered an "oven effect" and the "cooked" ashes left their fingerprint in a cavity, with just bones being left for many centuries, potentially suffering post-depositional processes before their discovery and infilling with plaster by archaeologists.

## Conclusions

For the first time the bone remains of fugitives from the Pompeii eruption have been directly analysed employing non-destructive analytical techniques (pXRF). The proposed approach behind this research consisted of comparing these results with those obtained on burned and buried bones employed as reference collections to evaluate the cast bones contamination by plaster. Then, based on data related to the most intact samples, the possible perimortem and post-mortem processes were discussed, cross-referencing chemical results with the anthropological and taphonomic study, and the volcano stratigraphic information found in the literature. The obtained results are clearly helpful in the reconstruction of the perimortem and post-mortem events concerning the history of these individuals and maybe, this study can shed light on the possible causes of death during the volcano eruption and create the prerequisites for a protocol to be applied to casts from Pompeii and other Vesuvian areas.

Our developed hypothesis for the cause of death is only applicable to the studied context. It is likely that the catastrophic eruption killed people in different ways and all the discussed

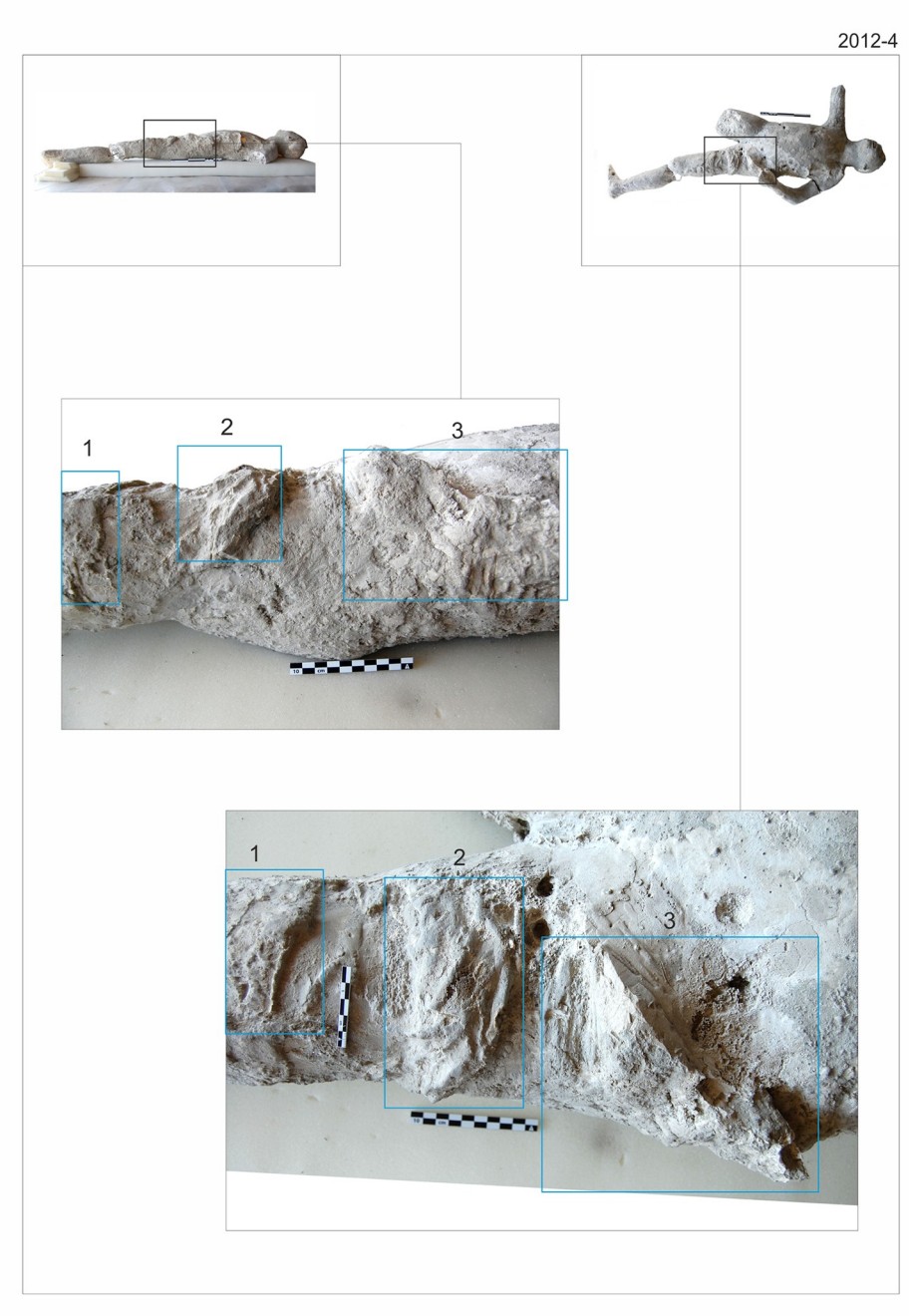

**Fig 12. Clothing details of the cast #57.**

scenarios should be taken into account. Generalizing and supporting a sole hypothesis of death becomes overly reductive.

Moreover, the use of chemical analyses on the cast skeletons is encouraged to assess the extent of post-depositional alteration and plaster contamination prior to any future biomineral or bimolecular analyses, in order to avoid misleading interpretations.

## Supporting information

**S1 File. Biological profile of the casts.**
(DOCX)

**S2 File. Sample description.** Results for pXRF analysis. Elemental concentrations for the studied samples.
(XLSX)

**S3 File. Other major and minor element graphs.**
(DOCX)

## Acknowledgments

The authors would like to thanks all the collaborators and students for their support during the 2019 Pompeii Archaeological Park fieldwork, and also all the public institutions that have contributed to the realization of this study.

Thanks to the professor Agustin Diez Castillo, the director of MAOVA museum of de Vall d' Albaida Albert Ribera and the PhD student Alessia Larini to provide the permit to access to some bone fragments from the Islamic necropolis of Colata (Montaverner), located in the Vall d'Albaida (Valencia, Spain).

Finally, we are especially grateful to the Editor and Reviewers for their work that has consistently improved the quality of the manuscript.

## Ethics statement

The authors have fulfilled all of the ethical requirements mandated by the Ethics council at the University of Valencia and at all the involved institutions including the Archaeological Park of Pompeii, the Sovrintendenza Capitolina ai Beni Culturali Direzione interventi su Edilizia Monumentale, Servizio Coordinamento Monumenti Antichi e Aree Archeologiche and the Museu Arqueòlogic d'Ontinyent Vall d'Albaida. Therefore this research is compatible with EU and international law and has been approved by the ethics committee of the University of Valencia.

The permits for all the studied materials have been regularly obtained and the authors ensure that conditions of the permits are met.

In case of casts of Pompeii (#57, #62, #58, #64, #54, #55) not sampling have been carried out and just in situ non-destructive measurements by pXRF have been obtained (C57.1–6; C62.1–6;C58.1–6; C34.1–6; C64.2–6; C54.1–6; C55.1–6). Regarding the burned bones fragments from the necropolis of Porta Nola (Pompeii, Italy), 21 samples corresponding to different individuals and type of bones have been sampled (C006e; R008eC; R014e; T027.1eb; U094e; C027.2e; F033e; T055; U058.1e; P058.2e; T071.2eb; D073e; T253e; R262e; S666e; R696e; F724b; S802e; C808e; C809e; C075e) and transported to carry out analyses in the laboratory of the department of Analytical Chemistry of the University of Valencia (Burjassot, Valencia, Spain). The permits to analyse the casts in situ (signed date: 9/08/2019) and the bone fragments (signed date: 19/08/2019; prot. N. 11496) were authorized by the director of the Pompeii Archaeological Park, Massimo Osanna.

About the burned bones fragments from the necropolis of Sepolcreto della via Ostiense (Rome, Italy), 24 samples corresponding to different individuals and type of bones (M162; M14; M17; M387; M458; M67; M62; M137; M72a; M161; M314; M80; M574; M43; M509; M418; M216; M172; M269; M313; M599; M27; M29; M30) and transported to carry out analyses in the laboratory of the department of Analytical Chemistry of the University of Valencia

(Burjassot, Valencia, Spain). The permits for bone fragments sampling were authorized by the head of service Maria Gabriella Cimino and the director Antonello Fatello from the Sovrinten- denza Capitolina ai Beni Culturali Direzione interventi su Edilizia Monumentale, Servizio Coordi- namento Monumenti Antichi e Aree Archeologiche (signed date: 11/09/2020; prot N. RI/21891).

Finally, the inhumated bones fragments from the necropolis of Colada (Montaverner, Valencia, Spain), 11 samples corresponding to different individuals and type of bones (IB01; IB02; IB03; IB04; IB05; IB06; IB07; IB08; IB09; IB11; IB12) were transported to carry out analy- ses at the laboratory of the department of Analytical Chemistry at the University of Valencia (Burjassot, Valencia, Spain). The permits for bone fragments sampling were authorized by the director of de Museu Arqueòlogic d'Ontinyent Vall d'Albaida, Agustí Ribera (signed date: 02/ 06/2020; exp 20200603).

All the samples details are available in the S2 File. Casts detailed description can be found in S1 File.

## Author Contributions

**Conceptualization:** Llorenç Alapont, Gianni Gallello, Marcos Martinón-Torres, Simon Chen- ery, Mirco Ramacciotti.

**Data curation:** Llorenç Alapont, Gianni Gallello, Simon Chenery, Mirco Ramacciotti, Ángel Morales Rubio, M. Luisa Cervera, Agustín Pastor.

**Formal analysis:** Llorenç Alapont, Gianni Gallello, Agustín Pastor.

**Funding acquisition:** Llorenç Alapont, Gianni Gallello, Valeria Amoretti, Ángel Morales Rubio, M. Luisa Cervera, Agustín Pastor.

**Investigation:** Llorenç Alapont, Gianni Gallello, Simon Chenery, Mirco Ramacciotti.

**Methodology:** Llorenç Alapont, Gianni Gallello, Marcos Martinón-Torres, Simon Chenery, Mirco Ramacciotti.

**Project administration:** Llorenç Alapont, Gianni Gallello, Massimo Osanna, Valeria Amo- retti, Agustín Pastor.

**Resources:** Llorenç Alapont, Gianni Gallello, Massimo Osanna, Valeria Amoretti, José Luis Jiménez, Ángel Morales Rubio, M. Luisa Cervera.

**Software:** Llorenç Alapont, Gianni Gallello, Mirco Ramacciotti.

**Supervision:** Llorenç Alapont, Gianni Gallello, Marcos Martinón-Torres, Massimo Osanna, Valeria Amoretti, José Luis Jiménez, Ángel Morales Rubio, M. Luisa Cervera, Agustín Pastor.

**Validation:** Llorenç Alapont, Gianni Gallello, Marcos Martinón-Torres, Massimo Osanna, Valeria Amoretti, José Luis Jiménez, Ángel Morales Rubio, M. Luisa Cervera, Agustín Pastor.

**Visualization:** Llorenç Alapont, Gianni Gallello, Marcos Martinón-Torres, Massimo Osanna, Simon Chenery, Mirco Ramacciotti, José Luis Jiménez, Ángel Morales Rubio, M. Luisa Cer- vera, Agustín Pastor.

**Writing – original draft:** Gianni Gallello.

**Writing – review & editing:** Llorenç Alapont, Gianni Gallello, Marcos Martinón-Torres, Mas- simo Osanna, Valeria Amoretti, Simon Chenery, Mirco Ramacciotti, José Luis Jiménez, Ángel Morales Rubio, M. Luisa Cervera, Agustín Pastor.

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
