## [Decision Letter · Decision Letter 0]

16 May 2023

PONE-D-23-12449The Casts of Pompeii: New insightsPLOS ONE

Dear Dr. Gallello,

Thank you for submitting your manuscript to PLOS ONE. After careful consideration, we feel that it has merit but does not fully meet PLOS ONE’s publication criteria as it currently stands. Therefore, we invite you to submit a revised version of the manuscript that addresses the points raised during the review process.

ACADEMIC EDITOR: 

Dear Dr. Gallello,

We appreciate you submitting your manuscript to PLOS ONE and thank you for giving us the opportunity to consider your work.

I have completed my evaluation of your manuscript, which has been reviewed by four highly qualified reviewers all of whom agree it is worth to be published in PLOS ONE. Nevertheless, they have suggested some changes that will help to improve the paper.

Therefore, I invite you to resubmit your manuscript after addressing the reviewers’ comments below. When revising your manuscript, please consider all issues mentioned in the reviewers' comments carefully: please, outline every change made in response to their comments and provide suitable rebuttals for any comments not addressed. Please, note that your revised submission may need to be re-reviewed.

On the other hand, the journal staff notified me that "Manuscripts reporting paleontology and archaeology research must adhere to our policies described at http://journals.plos.org/plosone/s/submission-guidelines#loc-paleontology-and-archaeology-research. Specifically, appropriate identification numbers for the human remains, specimens and/or samples should be provided, and the data used in the study should be publicly deposited or made accessible for replication of the study. If applicable, please ensure permission to conduct destructive sampling was obtained.”

Based on the above, in your manuscript, please provide additional information regarding the specimens used in your study. Ensure that you have reported specimen numbers and complete repository information, including museum name and geographic location.

For more information on PLOS ONE's requirements for paleontology and archaeology research, see https://journals.plos.org/plosone/s/submission-guidelines#loc-paleontology-and-archaeology-research.

PLOS ONE values your contribution and I look forward to receiving your revised manuscript.

Yours sincerely,

Dr. Olga Spekker

We look forward to receiving your revised manuscript.

Kind regards,

Olga Spekker, Ph.D.

Academic Editor

PLOS ONE

Journal Requirements:

2. In your manuscript, please provide additional information regarding the specimens used in your study. Ensure that you have reported human remain specimen numbers and complete repository information, including museum name and geographic location. 

For more information on PLOS ONE's requirements for paleontology and archeology research, see https://journals.plos.org/plosone/s/submission-guidelines#loc-paleontology-and-archaeology-research

Reviewers' comments:

Reviewer's Responses to Questions

**Comments to the Author**

1. Is the manuscript technically sound, and do the data support the conclusions?

Reviewer #1: Partly

Reviewer #2: Yes

Reviewer #3: Partly

Reviewer #4: Yes

2. Has the statistical analysis been performed appropriately and rigorously? 

Reviewer #1: Yes

Reviewer #2: Yes

Reviewer #3: Yes

Reviewer #4: Yes

3. Have the authors made all data underlying the findings in their manuscript fully available?

Reviewer #1: Yes

Reviewer #2: Yes

Reviewer #3: Yes

Reviewer #4: Yes

4. Is the manuscript presented in an intelligible fashion and written in standard English?

Reviewer #1: Yes

Reviewer #2: Yes

Reviewer #3: Yes

Reviewer #4: No

5. Review Comments to the Author

Reviewer #1: This review follows the thematic points provided by Plos One.

1. The study presents the results of original research.

Yes

2. Results reported have not been published elsewhere.

Correct - they have not

3. Experiments, statistics, and other analyses are performed to a high technical standard and are described in sufficient detail.

Partial. Bioarchaeological analysis – detail on what methods used for sex/age are needed. Currently it mostly states the sex, age result, and describes body position. If the profiling is obtained from other researchers, this needs citation.

pXRF - some clarity on method needed (see final paragraph below).

4. Conclusions are presented in an appropriate fashion and are supported by the data.

Yes – but more can be brought out on the elemental analysis

5. The article is presented in an intelligible fashion and is written in standard English.

Yes – I have added comments on some bits for clarification but it’s quite nit-picky – it’s well written.

6. The research meets all applicable standards for the ethics of experimentation and research integrity.

This research investigates a moment of trauma and death among the human population in Pompeii. It directly handles the material in question. Ethics statement said NA but I feel this requires some ethics statement.

7. The article adheres to appropriate reporting guidelines and community standards for data availability.

Yes – but the Cu data is missing from supplementary information.

This was an interesting article to read, thank you for sending it for review. It was concise and to the point, and presented some effective images and reading.

I have added detailed comments to the manuscript to read through – it’s mostly on some bits of clarity and should be quite quick to address.

However, I can’t ascertain what the impact of the pXRF work was. The research is framed around the bioarchaeological and elemental analysis bringing new insight but the bioarch work is quite routine on the Pompeii casts, and the results/discussion have limited content on what exactly the elemental analysis showed – what interpretation have you gathered from the data? What I can mostly see is that plaster and bone are different elemental compositions – which is expected as they’re entirely different materials. This will need strengthening and more depth in discussion to really bring the value out of your large dataset.

The pXRF method also needs some clarity as it currently poses a little risk. Did you scan samples once, in triplicate, or other? If once, this needs to be repeated, bones are highly variable. Did you achieve infinite thickness in the scans? How did you scan the bones, was it in direct contact with the cleaned bone surface or was it through the plaster? pXRF and XRF beams don’t penetrate that far into the material and will be interfered by the plaster.

Reviewer #2: The paper deals with the detail analysis of casts at Pompei with the aim of understanding the cause of death of people being impacted by the last phases of the Vesuvius eruption of 79 AD. The research is based mostly on bioarcheological data consisting of chemical measurements, by portable XRF, on bones and casts. Interpretation, namely asphyxia, is coherent with recent papers assessing the fate of people dying after the eruption based on volcanological interpretation. This multidisciplinary convergence enforces the interpretation of the natural event, which is useful also for assessing the hazard potential of devastating eruptions of Vesuvius. For these reasons I believe the paper deserves to be published on a high impact journal as PLOSONE.

Moderate revision is recommended, after consideration of the following main points

1) English needs a careful revision especially of the casts description

2) People escaping from Pompei after the first phase of the eruption were invested by the dilute pyroclastic density currents of the final phase which were caused by the eruptive column collapse. The passage of these currents lasted for several minutes and even though the temperature of the gas-ash mixture was not very high it cannot be breath for long, causing asphyxia. Therefore, asphyxia was caused by the passage of the pyroclastic density current itself, not by ash being suspended as resulting from the previous fallout of pumice of the first phase of the eruption. Afterward, the deposition from the current formed a layer of hot ash that had thermal impact on corpse of people that have already died for asphyxia and were lying on the ground. I do suggest authors to revise the paper in the discussion section as be more consistent with such a reconstruction of the eruptive events.

Regards

Pierfrancesco Dellino

Reviewer #3: There are essentially two components to your paper: the analysis of whether the plaster casts alter the chemical composition of the bone samples, as well as an interpretation on manner of death. The two topics (in my opinion) feel a bit disjointed, and having them both but in this paper seems a little forced.

I see nothing wrong with any of your conclusions, however they are not exactly ground breaking.

A few comments on spelling or where more clarification would be useful:

L 128 – use passive tense “only half the humeral shaft is available” rather than “we only have”

L 134 – I assume by “first half of femoral shaft” the proximal half is meant?

L 142 – I fail to see how the sutures or teeth give indications on sex determination?

L 180 – “carpals” not carpus; metacarpal not metacarpus

L 182 – humerus no humours

L 207 – one of the “previous” is redundant

L 331 – 334 – not sure I am following the rationale here. Also what “new authigenic phases” are referred to? A bit more information would be useful.

L 353 – 260 – this paragraph should be introduced in the intro lit review, the discussion should onl refer to concepts previously mentioned.

L 386 – lixivitation is not an English word.

Reviewer #4: Overall, this is a very interesting manuscript that examines anthropological, paleopathological, and chemical analysis of human bones and casts of Pompeii victims. The paper should be published and offers new insights into the final moments of the victim's lives and their deaths. The manuscript does need substantial editing and the PDF of the manuscript has been marked up with comments, suggestions, and other edits. Some statements are overly speculative, which is noted directly on the PDF. With some editing, this paper would make a nice contribution to the literature and to PLOS ONE.

6. PLOS authors have the option to publish the peer review history of their article (what does this mean?). If published, this will include your full peer review and any attached files.

Reviewer #1: No

Reviewer #2: **Yes: **Pierfrancesco Dellino

Reviewer #3: No

Reviewer #4: No

---

## [Author Response · Author response to Decision Letter 0]

30 Jun 2023

AUTHORS RESPONSE; 

Dear editor,

We are really grateful for giving us the possibility to revise the manuscript and answer the reviewer’s comments. We have worked hard to address all the points raised by the four referees and these have consistently improved the quality of our manuscript. Therefore, we really thank the reviewers for their work.

Reviewer #1 and Reviewer #4 comments and suggestions in the corresponding files together with the Reviewers #2 and #3 have been taken into account and reported in the new version of the manuscript in revision mode with our answers and comments.

Comments to the Author

1. Is the manuscript technically sound, and do the data support the conclusions?

Reviewer #1: Partly

Reviewer #2: Yes

Reviewer #3: Partly

Reviewer #4: Yes

2. Has the statistical analysis been performed appropriately and rigorously?

Reviewer #1: Yes

Reviewer #2: Yes

Reviewer #3: Yes

Reviewer #4: Yes

3. Have the authors made all data underlying the findings in their manuscript fully available?

Reviewer #1: Yes

Reviewer #2: Yes

Reviewer #3: Yes

Reviewer #4: Yes

4. Is the manuscript presented in an intelligible fashion and written in standard English?

Reviewer #1: Yes

Reviewer #2: Yes

Reviewer #3: Yes

Reviewer #4: No

5. Review Comments to the Author

Reviewer #1: This review follows the thematic points provided by Plos One.

 1. The study presents the results of original research.

Yes

2. Results reported have not been published elsewhere.

Correct - they have not

3. Experiments, statistics, and other analyses are performed to a high technical standard and are described in sufficient detail.

Partial. Bioarchaeological analysis – detail on what methods used for sex/age are needed. Currently it mostly states the sex, age result, and describes body position. If the profiling is obtained from other researchers, this needs citation.

pXRF - some clarity on method needed (see final paragraph below).

AUTHORS RESPONSE; We are grateful to the Reviewer #1 for the revision of our manuscript. The physical anthropological analysis in the casts was carried out directly in the field by one of the authors (Dr Llorence Alapont). The methodology employed to determine sex, age and all the anthropological aspects for the study of the 7 casts and the other reference collections follows the standardized parameters implemented by other authors. In the revised version of the manuscript, in methods section a new paragraph “Anthropological study” has been added and citations included as requested by Reviewer #1. About pXRF analysis the methodological aspects are clarified in the corresponding paragraph pXRF.

4. Conclusions are presented in an appropriate fashion and are supported by the data.

Yes – but more can be brought out on the elemental analysis

AUTHORS RESPONSE; Thanks, in conclusion sections some lines have been added to enhance the elemental results.

5. The article is presented in an intelligible fashion and is written in standard English.

Yes – I have added comments on some bits for clarification but it’s quite nit-picky – it’s well written.

AUTHORS RESPONSE; Thanks, in the new version of the manuscript, we have taken into consideration the language revision.

6. The research meets all applicable standards for the ethics of experimentation and research integrity.

This research investigates a moment of trauma and death among the human population in Pompeii. It directly handles the material in question. Ethics statement said NA but I feel this requires some ethics statement.

AUTHORS RESPONSE; Thanks. In ethics section “Ethics statement” we have added that no samples have been collected from the casts and direct non-destructive analysis were carried.

In all the case including burned Pompeii bones from pre Vesuvius eruption and bones from Islamic population the permit have been regularly obtained following the Spanish, Italian, European and International rules and regulations.

7. The article adheres to appropriate reporting guidelines and community standards for data availability.

Yes – but the Cu data is missing from supplementary information.

AUTHORS RESPONSE; Thanks for your comment, copper data were not shown because they were below the limit of detection for all the analysed materials and remains. Therefore, Cu has been removed from the manuscript revised version.

This was an interesting article to read, thank you for sending it for review. It was concise and to the point, and presented some effective images and reading.

I have added detailed comments to the manuscript to read through – it’s mostly on some bits of clarity and should be quite quick to address.

AUTHORS RESPONSE; Thanks for the Reviewer #1 positive comments. We believe that this study is pushing forward to develop a non-destructive methodological approach for extraordinary remains like the casts of Pompeii. Consequently, we had the possibility to determine the chemical profiles of bones and plaster of some the casts for the first time, cross-referencing the data with the anthropological research. This allowed us to better understand perimortem and post-mortem processes.

However, I can’t ascertain what the impact of the pXRF work was. The research is framed around the bioarchaeological and elemental analysis bringing new insight but the bioarch work is quite routine on the Pompeii casts, and the results/discussion have limited content on what exactly the elemental analysis showed – what interpretation have you gathered from the data? What I can mostly see is that plaster and bone are different elemental compositions – which is expected as they’re entirely different materials. This will need strengthening and more depth in discussion to really bring the value out of your large dataset.

AUTHORS RESPONSE; Thanks for the Reviewer #1 comments that give us the opportunity to clarify the role of the chemical information obtained by pXRF analysis in inhumated burned remains and plaster. Once obtained the results the first steps were to develop an approach to verify if the plaster is affecting the chemical compounds of the bones. To do it, we statistically compare the elemental contents of inhumated bones burned bones and casts bones with casts plaster to select the cast bones minimally affected by the plaster contamination. The next step was to compare inhumated bones with Pompeii pre Vesuvian and Roma Ostiense burned bones collections, where we could observe chemical differences between these two big groups, that we explain with the thermal impact in the structure of the bones that causes the loss of some major elements during post-depositional processes, as confirmed by previous studies. As mentioned, “probably the thermal impact together with post-depositional processes induced the leaching of carbonates and phosphates and this could explain the lower calcium and phosphorous concentrations in burned bones [25, 26]”. Once we had made sure that these observations were “true” by running statistical analysis of the obtained chemical information, we could compare the Casts bones with the inhumated and burned bones, observing that the composition of the stated casts was similar to those of the burned bones. That gives us an important input to develop hypothesis about the post-mortem exposure of the Pompeii remains to quite high temperature being their death caused by the asphyxia as showed by anthropological studies and the preservation of garments. 

Following the Reviewer #1 comments a paragraph “The elemental data” has been added in the discussion section to further enhance the important information provided by pXRF analyses.

The pXRF method also needs some clarity as it currently poses a little risk. Did you scan samples once, in triplicate, or other? If once, this needs to be repeated, bones are highly variable. Did you achieve infinite thickness in the scans? How did you scan the bones, was it in direct contact with the cleaned bone surface or was it through the plaster? pXRF and XRF beams don’t penetrate that far into the material and will be interfered by the plaster.

AUTHORS RESPONSE; Thanks for the comments, all the materials and remains were measured in triplicate in the same point and the new version of the manuscript this is now specified and in Supplementary Information (S2 File) the standard deviation has been added to observe the measurements variability. For all the seven studied casts, the measurements were carried out in the exposed bones, not in the bones inside the plaster. The casts presented with some bone remains out of the plaster although partially fixed in the plaster surface. The exposed parts of the bones were carefully cleaned in their surface to remove the dust. Once cleaned the analytical zone of the pXRF was pressed against the bone surface and measured. In the same way the plaster surface was measured to control bones contamination. To make sure that the infinite thickness is met just the most compact and thickest remains and plaster parts were measured. Furthermore some previous tests in similar bones and plaster employing pXRF were carried out in our labs under controlled conditions to evaluate the reliability and strength of the analytical method. 

All the Reviewer #1 comments reported in the manuscript revised mode version are answered as comments in the attached document.

Reviewer #2: The paper deals with the detail analysis of casts at Pompei with the aim of understanding the cause of death of people being impacted by the last phases of the Vesuvius eruption of 79 AD. The research is based mostly on bioarcheological data consisting of chemical measurements, by portable XRF, on bones and casts. Interpretation, namely asphyxia, is coherent with recent papers assessing the fate of people dying after the eruption based on volcanological interpretation. This multidisciplinary convergence enforces the interpretation of the natural event, which is useful also for assessing the hazard potential of devastating eruptions of Vesuvius. For these reasons I believe the paper deserves to be published on a high impact journal as PLOSONE.

AUTHORS RESPONSE; We are really grateful to Reviewer #2 for his positive comments. Therefore this study shows a multidisciplinary approach cross-referencing pXRF chemical data with anthropological research and previous published research, supporting a death for asphyxia of the studied individuals and adding information about perimortem and post-mortem processes. Finally we developed a non-destructive methodological approach for evaluating plaster contamination to discard plaster contaminated bones and avoid misleading interpretations about bone casts original chemical contents. 

Moderate revision is recommended, after consideration of the following main points

1) English needs a careful revision especially of the casts description

AUTHORS RESPONSE; Thanks, in the new version of the manuscript the language has been revised, especially in the Results section “Bioarchaeological analysis on casts”.

2) People escaping from Pompei after the first phase of the eruption were invested by the dilute pyroclastic density currents of the final phase which were caused by the eruptive column collapse. The passage of these currents lasted for several minutes and even though the temperature of the gas-ash mixture was not very high it cannot be breath for long, causing asphyxia. Therefore, asphyxia was caused by the passage of the pyroclastic density current itself, not by ash being suspended as resulting from the previous fallout of pumice of the first phase of the eruption. Afterward, the deposition from the current formed a layer of hot ash that had thermal impact on corpse of people that have already died for asphyxia and were lying on the ground. I do suggest authors to revise the paper in the discussion section as be more consistent with such a reconstruction of the eruptive events.

AUTHORS RESPONSE; We are really grateful to Professor Dellino for his very clarifying description of the eruption phases, in the Discussion section in the new paragraph “The cause of the death and the post-mortem process” of the new version of the manuscript the description of the events has been improved following the reviewer comments.

Regards

Pierfrancesco Dellino

Reviewer #3: There are essentially two components to your paper: the analysis of whether the plaster casts alter the chemical composition of the bone samples, as well as an interpretation on manner of death. The two topics (in my opinion) feel a bit disjointed, and having them both but in this paper seems a little forced.

AUTHORS RESPONSE; We are grateful to Reviewer #3 for her/his comments. It is important to highlight that the two mentioned topics are strictly connected to support our discussion and conclusions about the perimortem and post-mortem processes. The first important step was to develop an approach to verify if the plaster is contaminating and affecting the chemical compounds of the bones. To do this, we designed a reference set of results analysing and statistically processing the elemental contents of inhumated bones, burned bones and casts bones with casts plaster to select the cast bones minimally affected by the plaster contamination. The next step was to compare inhumated bones with Pompeii pre Vesuvian and Roma Ostiense burned bones collections, where we could observe chemical differences between these two big groups that we explain with the thermal impact in the structure of the bones that causes the loss of some major elements during post-depositional processes as confirmed by previous studies. Once we had made sure that these observations are “true” by running statistical analysis of the obtained chemical information, we could compare the cast bones with the reference set of inhumated and burned, observing that the composition of the studied casts were similar to those of the burned bones. That gives us important information to develop hypothesis about the postmortem exposure of the Pompeii remains to quite high temperature being their death caused by the asphyxia as showed by anthropological studies and the preservation of garments. Finally all supported by the volcano stratigraphic literature.

I see nothing wrong with any of your conclusions, however they are not exactly ground breaking.

AUTHORS RESPONSE; This study shows a multidisciplinary approach cross-referencing pXRF chemical data with anthropological research and geological previous published research, supporting a death for asphyxia of the studied individuals and adding information about perimortem and post-mortem processes. Finally we developed a non-destructive methodological approach for evaluating plaster contamination to discard plaster contaminated bones and avoid misleading interpretations about bone casts original chemical contents. Therefore from a broader point of view here we present an innovative approach to be applied to casts from Pompeii and other Vesuvius areas shading light on the volcano eruption deaths and suggesting for the first time a protocol to be applied for the study of the casts.

A few comments on spelling or where more clarification would be useful:

L 128 – use passive tense “only half the humeral shaft is available” rather than “we only have” 

AUTHORS RESPONSE; Thanks for correcting. 

L 134 – I assume by “first half of femoral shaft” the proximal half is meant?

AUTHORS RESPONSE; Yes. Thanks this was corrected.

L 142 – I fail to see how the sutures or teeth give indications on sex determination?

AUTHORS RESPONSE; Thanks for the correction. This is a mistake it was corrected. 

L 180 – “carpals” not carpus; metacarpal not metacarpus

AUTHORS RESPONSE; Corrected in the new version of the text.

L 182 – humerus no humours

AUTHORS RESPONSE; Thank you for correcting. 

L 207 – one of the “previous” is redundant

AUTHORS RESPONSE; Thanks this has been corrected.

L 331 – 334 – not sure I am following the rationale here. Also what “new authigenic phases” are referred to? A bit more information would be useful.

AUTHORS RESPONSE; Thanks, further information has been added to clarify this issue. In discussion section of the new version of the manuscript “The mechanisms behind these processes were further explained by Nielsen et al. [36] being bone surface apatite heavily reactive, after death the biologically formed minerals become unstable and susceptible to the burial environment, therefore Ca2+ and P4-3 ions are expose and can be totally or partially replaced by the ions of other elements.”

L 353 – 260 – this paragraph should be introduced in the intro lit review, the discussion should onl refer to concepts previously mentioned.

We agree with Reviewer #3, as suggested this issue was introduced with a sentence in the introduction section.

L 386 – lixivitation is not an English word.

AUTHORS RESPONSE; Thanks for correcting. We refer at the word lixiviation as the process of lixiviating “the extraction of soluble components of a solid mixture by percolating a solvent through it”. However the word has been substituted with “leaching”. 

Reviewer #4: Overall, this is a very interesting manuscript that examines anthropological, paleopathological, and chemical analysis of human bones and casts of Pompeii victims. The paper should be published and offers new insights into the final moments of the victim's lives and their deaths. The manuscript does need substantial editing and the PDF of the manuscript has been marked up with comments, suggestions, and other edits. Some statements are overly speculative, which is noted directly on the PDF. With some editing, this paper would make a nice contribution to the literature and to PLOS ONE. 

AUTHORS RESPONSE; We are really grateful for the Reviewer #4 positive response. All the comments and suggestions have been taken into accounts and reported in the new version of the manuscript revised mode with our answers as comments in the same document.

6. PLOS authors have the option to publish the peer review history of their article (what does this mean?). If published, this will include your full peer review and any attached files.

Do you want your identity to be public for this peer review? For information about this choice, including consent withdrawal, please see our Privacy Policy.

Reviewer #1: No

Reviewer #2: Yes: Pierfrancesco Dellino

Reviewer #3: No

Reviewer #4: No

---

## [Decision Letter · Decision Letter 1]

14 Jul 2023

PONE-D-23-12449R1The Casts of Pompeii: post-depositional methodological insightsPLOS ONE

Dear Dr. Gallello,

Thank you for submitting your manuscript to PLOS ONE. After careful consideration, we feel that it has merit but does not fully meet PLOS ONE’s publication criteria as it currently stands. Therefore, we invite you to submit a revised version of the manuscript that addresses the points raised during the review process.

ACADEMIC EDITOR:

Dear Dr. Gallello,

We appreciate you submitting your manuscript to PLOS ONE and thank you for giving us the opportunity to consider your work.

I have completed my evaluation of your revised manuscript, which has been reviewed by two highly qualified reviewers all of whom agree it is worth to be published in PLOS ONE. Nevertheless, one of the reviewers suggested some minor changes and I also have some minor comments that should be addressed before your manuscript would be accepted.

Based on the above, I invite you to resubmit your manuscript after amending it.

PLOS ONE values your contribution and I look forward to receiving your revised manuscript.

Yours sincerely,

Dr. Olga Spekker

My comments:

Line 57 – “also called” instead of “also called also”

Line 66 – “the pyroclastic” instead of “that the pyroclastic”

Line 68 – “then” instead of “than”

Line 85 – “Roman” instead of “roman”

Line 89 – “In total” instead of “In resume”

Line 107 – “methods” instead of “method”

Line 116 – “was” instead of “were”

Line 122 – “demonstrated that” instead of “demonstrated”

Line 182 – “detailed description of the studied casts” instead of “the studied casts detailed description”

Line 188 – “presents hypoplasia” instead of “shows hypoplasia defects presents hypoplasia”

Line 189 – “an enthesophyte” instead of “an enthesophytes”

Lines 198–199 – “Except for Ca and P and ratios reported …” – it should be rephrased

Line 204 – “for these elements” instead of “for both these elements”

Lines 219 and 227 – “radius” instead of “radio”

Line 245 – “For, PC2” – the comma is not necessary

Line 277 – “other kind of useful” instead of “other kind useful”

Line 308 – “died of asphyxia” instead of “died for asphyxia”

We look forward to receiving your revised manuscript.

Kind regards,

Olga Spekker, Ph.D.

Academic Editor

PLOS ONE

Journal Requirements:

Reviewers' comments:

Reviewer's Responses to Questions

**Comments to the Author**

1. If the authors have adequately addressed your comments raised in a previous round of review and you feel that this manuscript is now acceptable for publication, you may indicate that here to bypass the “Comments to the Author” section, enter your conflict of interest statement in the “Confidential to Editor” section, and submit your "Accept" recommendation.

Reviewer #2: All comments have been addressed

Reviewer #4: All comments have been addressed

2. Is the manuscript technically sound, and do the data support the conclusions?

Reviewer #2: (No Response)

Reviewer #4: Yes

3. Has the statistical analysis been performed appropriately and rigorously? 

Reviewer #2: (No Response)

Reviewer #4: Yes

4. Have the authors made all data underlying the findings in their manuscript fully available?

Reviewer #2: (No Response)

Reviewer #4: Yes

5. Is the manuscript presented in an intelligible fashion and written in standard English?

Reviewer #2: (No Response)

Reviewer #4: Yes

6. Review Comments to the Author

Reviewer #2: (No Response)

Reviewer #4: In this second review, I believe the author team has addressed the reviewer comments. The paper should be acceptable for PLOS ONE and will make a nice contribution to the literature on the use of archaeometric methods for studying the victims of Pompeii.

I only have a few minor edits:

Page 3, line 72: change "operated" to "conducted"

Page 3, line 85: Should roman be capitalized ("Roman")

Page 3, line 89: change "In resume" to "In total"

Page 4, line 99: "calcaneus and heel" are the same bone. Remove "and heel"

Page 4, line 104: change "were based" to "was based"

Page 4, line 117: change "materials" to "material"

7. PLOS authors have the option to publish the peer review history of their article (what does this mean?). If published, this will include your full peer review and any attached files.

Reviewer #2: **Yes: **pierfrancesco dellino

Reviewer #4: No

---

## [Author Response · Author response to Decision Letter 1]

16 Jul 2023

PONE-D-23-12449R1

The Casts of Pompeii: post-depositional methodological insights

PLOS ONE

Dear Dr. Gallello,

Thank you for submitting your manuscript to PLOS ONE. After careful consideration, we feel that it has merit but does not fully meet PLOS ONE’s publication criteria as it currently stands. Therefore, we invite you to submit a revised version of the manuscript that addresses the points raised during the review process.

ACADEMIC EDITOR:

Dear Dr. Gallello,

We appreciate you submitting your manuscript to PLOS ONE and thank you for giving us the opportunity to consider your work.

I have completed my evaluation of your revised manuscript, which has been reviewed by two highly qualified reviewers all of whom agree it is worth to be published in PLOS ONE. Nevertheless, one of the reviewers suggested some minor changes and I also have some minor comments that should be addressed before your manuscript would be accepted.

Based on the above, I invite you to resubmit your manuscript after amending it.

PLOS ONE values your contribution and I look forward to receiving your revised manuscript.

Yours sincerely,

Dr. Olga Spekker

-AUTHORS RESPONSE: Dear editor we are very grateful for your comments and suggestions, all the recommended changes have been amended. Furthermore the text has been double checked and revised to guarantee the quality of the manuscript and avoid as far as we can typos.

My comments:

Line 57 – “also called” instead of “also called also”

Line 66 – “the pyroclastic” instead of “that the pyroclastic”

Line 68 – “then” instead of “than”

Line 85 – “Roman” instead of “roman”

Line 89 – “In total” instead of “In resume”

Line 107 – “methods” instead of “method”

Line 116 – “was” instead of “were”

Line 122 – “demonstrated that” instead of “demonstrated”

Line 182 – “detailed description of the studied casts” instead of “the studied casts detailed description”

Line 188 – “presents hypoplasia” instead of “shows hypoplasia defects presents hypoplasia”

Line 189 – “an enthesophyte” instead of “an enthesophytes”

Lines 198–199 – “Except for Ca and P and ratios reported …” – it should be rephrased

Line 204 – “for these elements” instead of “for both these elements”

Lines 219 and 227 – “radius” instead of “radio”

Line 245 – “For, PC2” – the comma is not necessary

Line 277 – “other kind of useful” instead of “other kind useful”

Line 308 – “died of asphyxia” instead of “died for asphyxia”

-AUTHORS RESPONSE: Thanks, the suggested changes have been made. Furthermore the word radius has been correctly written also in the Supporting Information S3.

Journal Requirements:

-AUTHORS RESPONSE: Thanks, the reference list has been revised to ensure that it is complete and correct. As far as we know retracted papers have not been cited.

Reviewers' comments:

Reviewer's Responses to Questions

Comments to the Author

1. If the authors have adequately addressed your comments raised in a previous round of review and you feel that this manuscript is now acceptable for publication, you may indicate that here to bypass the “Comments to the Author” section, enter your conflict of interest statement in the “Confidential to Editor” section, and submit your "Accept" recommendation.

Reviewer #2: All comments have been addressed

Reviewer #4: All comments have been addressed

2. Is the manuscript technically sound, and do the data support the conclusions?

Reviewer #2: (No Response)

Reviewer #4: Yes

3. Has the statistical analysis been performed appropriately and rigorously?

Reviewer #2: (No Response)

Reviewer #4: Yes

4. Have the authors made all data underlying the findings in their manuscript fully available?

Reviewer #2: (No Response)

Reviewer #4: Yes

5. Is the manuscript presented in an intelligible fashion and written in standard English?

Reviewer #2: (No Response)

Reviewer #4: Yes

6. Review Comments to the Author

Reviewer #2: (No Response)

Reviewer #4: In this second review, I believe the author team has addressed the reviewer comments. The paper should be acceptable for PLOS ONE and will make a nice contribution to the literature on the use of archaeometric methods for studying the victims of Pompeii.

I only have a few minor edits:

Page 3, line 72: change "operated" to "conducted"

Page 3, line 85: Should roman be capitalized ("Roman")

Page 3, line 89: change "In resume" to "In total"

Page 4, line 99: "calcaneus and heel" are the same bone. Remove "and heel"

Page 4, line 104: change "were based" to "was based"

Page 4, line 117: change "materials" to "material"

-AUTHORS RESPONSE: The Reviewer #4 minor edits have been completed as suggested. Excepting for line 104 (change "were based" to "was based) and line 117 (change "materials" to "material): disagree as plural reference and not changed.

7. PLOS authors have the option to publish the peer review history of their article (what does this mean?). If published, this will include your full peer review and any attached files.

Do you want your identity to be public for this peer review? For information about this choice, including consent withdrawal, please see our Privacy Policy.

Reviewer #2: Yes: pierfrancesco dellino

Reviewer #4: No

-AUTHORS RESPONSE: We are especially grateful to the Editor and Reviewers for their work that, certainly, has consistently improved the quality of the manuscript. A sentence to thanks the Reviewers work has been added in “Acknowledgments” section.

---

## [Editor Report · Decision Letter 2]

18 Jul 2023

The Casts of Pompeii: post-depositional methodological insights

PONE-D-23-12449R2

Dear Dr. Gallello,

We’re pleased to inform you that your manuscript has been judged scientifically suitable for publication and will be formally accepted for publication once it meets all outstanding technical requirements.

Kind regards,

Olga Spekker, Ph.D.

Academic Editor

PLOS ONE
---

## [Editor Report · Acceptance letter]

27 Jul 2023

PONE-D-23-12449R2 

The Casts of Pompeii: post-depositional methodological insights 

Dear Dr. Gallello:

I'm pleased to inform you that your manuscript has been deemed suitable for publication in PLOS ONE. Congratulations! Your manuscript is now with our production department. 

Kind regards, 

on behalf of

Dr. Olga Spekker 

Academic Editor

PLOS ONE